# Proposal for the Inclusion of Tobacco Use in Suicide Risk Scales: Results of a Meta-Analysis

**DOI:** 10.3390/ijerph18116103

**Published:** 2021-06-05

**Authors:** Iván Echeverria, Miriam Cotaina, Antonio Jovani, Rafael Mora, Gonzalo Haro, Ana Benito

**Affiliations:** 1TXP Research Group, Universidad Cardenal Herrera-CEU, CEU Universities, 12006 Castellón, Spain; gomechiva@alumnos.uchceu.es (I.E.); miriam.cotaina@gmail.com (M.C.); antonio.jovani@hotmail.com (A.J.); gonzalo.haro@uchceu.es (G.H.); 2Department of Mental Health, Consorcio Hospitalario Provincial de Castellón, 12002 Castellón, Spain; rafael.mora@hospitalprovincial.es; 3Torrente Mental Health Unit, Hospital General de Valencia, 46900 Torrente, Spain

**Keywords:** suicide, suicidal behaviours, smoking, tobacco, nicotine, prospective, meta-analysis

## Abstract

There is an association between smoking and suicide, even though the direction and nature of this relationship remains controversial. This meta-analysis aimed to evaluate the association between smoking and suicidal behaviours (ideation, planning, suicide attempts, and death by suicide). On 24 August 2020, we searched the PubMed, Cochrane library, Scopus, Web of Science, TRIP, and SCIENCE DIRECT databases for relevant articles on this topic. Twenty prospective cohort studies involving 2,457,864 participants were included in this meta-analysis. Compared with never smokers, former and current smokers had an increased risk of death by suicide (relative risk [RR] = 1.31; 95% CI [1.13, 1.52] and RR = 2.41; 95% CI [2.08, 2.80], respectively), ideation (RR = 1.35; 95% CI [1.31, 1.39] and RR = 1.84; 95% CI [1.21, 2.78]), and attempted suicide (RR = 1.27; 95% CI [0.56, 2.87] and RR = 1.71; 95% CI [0.73, 3.97]). Moreover, compared to never smokers, current smoker women (RR = 2.51; 95% CI [2.06–3.04] had an increased risk of taking their own life (*Q* = 13,591.53; *p* < 0.001) than current smoker men (RR = 2.06; 95% CI [1.62–2.62]. Furthermore, smoking exposure (former and current smokers) was associated with a 1.74-fold increased risk (95% CI [1.54, 1.96]) of suicidal behaviour (death by suicide, ideation, planning, or attempts). Thus, because of the prospective relationship between smoking and suicidal behaviours, smoking should be included in suicide risk scales as a useful and easy item to evaluate suicide risk.

## 1. Introduction

When a person decides to attempt suicide, their whole environment falls into disarray. Each year, around 800,000 people take their own life globally [1]. Suicide is the leading cause of external death in countries such as Spain, where its incidence is well above that of deaths caused by traffic accidents [2]. Furthermore, for each death by suicide, there are more than 20 attempted suicides [1]. Despite these figures, the social stigma associated with suicide [3] and taboos and silence that surround it for fear of the ‘contagion effect’ [4], may be one reason why it is not prioritized as an essential public health problem. For this reason, we believe that the creation of prevention measures is of vital importance.

In contrast, in 2018, a total of 1337 million people worldwide were smokers [5], despite efforts by world governments to curb this habit. Considering that numerous studies suggest that tobacco use increases suicide risk [6,7,8,9], further research is warranted to deepen our scientific understanding of this relationship. Many of the risk factors for suicide are also risk factors for being a smoker: being young, non-white, lower income, less education, unmarried, unemployed, less religious, anxiety, depression, psychoses, substance use problems, low self-esteem, risk taking, having a serious physical illness, impulsivity, aggression, and antisocial personality [7]. This may allow tobacco use to be included in the suicide risk assessment protocols, and treatment of this addiction could be considered a social health intervention for implementation alongside other suicide prevention measures. However, the main suicide risk assessment instruments, such as the SAD PERSONS scale [10], do not currently include smoking as an independent predictive factor.

Previous meta-analyses conclude that smoking is associated with a higher risk of completed suicide [11]. However, we do not know how smoking could influence the rest of the suicidal spectrum behaviours, since we have not found any prospective meta-analyses that analyse each of these behaviours separately. In the case of concluding that such association exists, we would be closer to considering smoking as a modifiable risk factor for suicide.

Therefore, the objective of this present study was to carry out a meta-analysis (MA) of prospective cohort studies that assessed the magnitude of the relationship between tobacco use and the suicidal spectrum (autolytic ideation, suicidal plan, suicide attempt, and death by suicide). Previous suicide ideation is the most robust predictor of later ideation. Prior non-suicidal self-injuries or suicide attempts confer the most risk for later suicide attempts and previous suicide attempts and suicide ideation are among the strongest predictors of suicide death [7]. Our hypothesis was that the risk of implementing suicidal behaviours would follow the following order: smokers > ex-smokers > non-smokers. In contrast to previous studies [11,12], we considered every dimension of the suicidal spectrum and included studies from all over the world, without excluding populations with mental illnesses, such as substance-related disorders.

## 2. Materials and Methods

### 2.1. Protocol and Registration

This report was prepared according to the PRISMA guidelines for reporting in systematic reviews and meta-analyses [13]. The protocol was registered with the Prospero Centre for Reviews and Dissemination on 13 July 2020 (CRD42020197569), available at the following address: www.crd.york.ac.uk/PROSPERO/display_record.php?ID=CRD42020197569 (accessed on 4 June 2021).

### 2.2. Search Strategy

We searched for relevant studies in the PubMed, Cochrane library, Scopus, Web of Science, TRIP, and SCIENCE DIRECT databases on 24 August 2020. The search terms were ‘suicide OR suicidality OR suicidal OR self-inflicted death OR completed suicide’ AND ‘smoke OR smoking OR tobacco OR cigarette OR cigar OR nicotine OR tobacco use disorders OR electronic cigarette OR smokeless tobacco’ AND ‘prospective OR follow OR longitudinal OR cohort OR risk ratio OR hazard ratio OR survival analysis OR follow up OR relative risk’. We also reviewed the list of citations included in these articles, reviews, and meta-analyses.

### 2.3. Inclusion and Exclusion Criteria

The following inclusion criteria were applied: (1) prospective cohort design; (2) the exposure of interest was the smoking habit itself; (3) the participants were classified as non-smokers (NS), former smokers (FS), or current smokers (CS); (4) outcomes were suicidal ideation (thoughts about committing suicide), suicidal plans (making a plan for committing suicide), suicidal attempts (attempting suicide), and/or death by suicide; (5) original papers reporting relative risks (RRs) or hazard ratios (HRs) and their 95% confidence intervals (CIs) of suicidal ideation, plans, attempts, and/or deaths by suicide (or data to calculate these) were provided.

The exclusion criteria were as follows: (1) case–control, cross-sectional, or retrospective designs; (2) did not distinguish between current and former smokers; (3) the exposure was the age of initiation of smoking or high versus low smokers were compared; (4) mixed outcomes (suicidal behaviour, accidents, and violence) were reported, and the risk of suicidal behaviour could not be separated from among these data.

Prospective cohort studies in any type of population were used in this MA, irrespective of language, publication data, nationality, race, age, and gender.

### 2.4. Data Extraction

The articles assessed for eligibility were divided into two parts; two authors independently extracted the information from each half (a total of four researchers) by using a standardised form and resolving any disagreements though discussion with the other two authors until a consensus was achieved. The variables extracted from the studies were author names, year, country, language, study population, sample size, sample age, gender, follow-up duration, study quality evaluated through the Newcastle–Ottawa Scale (NOS; [14]), smoking exposure (NS, FS, and CS), suicidal outcomes (ideation, plans, attempts, and/or death), and RRs or HRs and related 95% CIs (preferably with the most adjusted factors). When these data were not provided directly, they were calculated using the data provided.

### 2.5. Summary Measures

We used the RR and their 95% CIs to state the association between smoking and suicidal behaviours and employed a random-effects model. We performed the statistical analyses using Epidat 3.1 software (Xunta de Galicia, A Coruña, Spain; Pan American Health Organization (WHO), Washington, DC, USA). Because this program was designed in Spain, in its outputs the decimals are expressed with commas. Therefore, in Figures 2–5, commas are to be interpreted as decimal points.

### 2.6. Quality Assessment

The NOS [14] was used to evaluate the methodological quality of the studies included. This scale allocates a maximum of nine stars to the domains: selection, comparability, exposure, and outcome of studies.

### 2.7. Heterogeneity and Publication Bias

Heterogeneity was evaluated using DerSimonian–Laird *Q* tests with Galbraith graphics. Possible publication bias was examined by employing Egger and Begg tests with funnel plots.

### 2.8. Sensitivity and Subgroup Analysis

We performed a sensitivity analysis by repeating the MA the same number of times as selected studies (*N* = 20), each time omitting one study and combining all the remaining one and plotting influence graphs. We decided whether we needed to analyse subgroups of studies by performing heterogeneity and sensitivity analyses.

## 3. Results

### 3.1. Studies Included

Our initial search returned 3800 articles. We eventually included 20 articles in this MA; Figure 1 shows the evaluation process we followed to select these studies.

Of the 20 selected articles, 15 examined deaths by suicide, 3 looked at suicidal ideation, and another 3 investigated suicide attempts. Although the results obtained regarding suicidal ideation and suicide attempts were unreliable because we found very few studies examining these areas, we included them for informational purposes. We only found one article that had studied suicide plans (FS RR = 1.20, 95% CI [0.60, 2.40]; CS RR = 1.50, 95% CI [0.90, 2.80]) and so the MA could not be carried out for this part of the spectrum. An MA was carried out for the relationship between exposure to tobacco (FE and CE) and all suicidal behaviours (ideation, plans, attempts, and death). Finally, to assess whether there was a difference between male and female smokers, the studies that presented gender data were selected and a separate MA was performed for each sex (6 studies for women and 9 studies for men). Table 1 shows the characteristics of the studies included in these meta-analyses [15,16,17,18,19,20,21,22,23,24,25,26,27,28,29,30,31,32,33,34]; all included studies were in English.

### 3.2. Smoking and Risk of Death by Suicide

The risk of death by suicide versus NS was 1.31 (95% CI [1.13, 1.52]) for FS and 2.41 (95% CI [2.08, 2.80]) for CS (Figure 2). The *Q* index was 19.12 (*p* = 0.160) in FS and 27.20 (*p* = 0.018) in CS.

Compared with NS, the risk of death by suicide for CS women was 2.51 (95% CI [2.06, 3.04]) and 2.06 (95% CI [1.62, 2.62]) for CS men (Figure 3). Women were significantly more at risk than men (*Q* = 13,591.53; *p* < 0.001); in women *Q* = 2.00 (*p* = 0.848) and in men *Q* = 21.54 (*p* = 0.005).

Galbraith graphics are shown in Appendix A.

In FS, both the *Q* index and the Galbraith graph indicate the absence of heterogeneity. In CS, the *Q* index indicates the presence of heterogeneity, but in the graph, it can be seen that only Iwasaki (2005) contributes to it. In women, both the *Q* index and the Galbraith graph indicate absence of heterogeneity. In men, the *Q* index indicates heterogeneity, but in the graph, it can be seen that Miller (2000b) is the one that contributes the most to this heterogeneity.

### 3.3. Smoking and Risk of Suicide Ideation

The risk of suicide ideation versus NS was 1.35 (95% CI [1.31, 1.39]) for FS and 1.84 (95% CI [1.21, 2.78]) for CS (Appendix A). The *Q* index was 1.87 (*p* = 0.391) in FS and 8.11 (*p* = 0.017) in CS. Galbraith graphics are shown in Appendix A.

In FS, both the *Q* index and the Galbraith graph indicate the absence of heterogeneity. In CS, the *Q* index indicates the presence of heterogeneity, but in the graph it can be seen that such heterogeneity is minimal.

### 3.4. Smoking and Risk of Suicide Attempts

The risk of suicide attempts versus NS was 1.27 (95% CI [0.56, 2.87]) for FS and 1.71 (95% CI [0.73, 3.97]) for CS (Appendix A). The *Q* index was 7.55 (*p* = 0.022) in FS and 22.13 (*p* < 0.001) in CS; Galbraith graphics are shown in Appendix A.

In FS, *Q* index indicates heterogeneity, but the Galbraith graph indicates the absence of heterogeneity. In CS, both the *Q* index and the graph indicate the presence of heterogeneity.

### 3.5. Smoking and Risk of Suicidal Behaviours

The risk of suicidal behaviours (ideation, plans, attempts, and death) in people exposed to smoking (FS and CS) was 1.74 (95% CI [1.54, 1.96]; Figure 4). The *Q* index was 645.22 (*p* < 0.001) and Galbraith graphs are shown in Appendix A. Both the *Q* index and the graph indicate the presence of heterogeneity.

### 3.6. Publication Bias

Regarding deaths by suicide, in FS the Berg test was *Z* = 0.79 (*p* = 0.428) and *t* = −0.13 (*p* = 0.895) for the Egger test, while in CS *Z* = 0.39 (*p* = 0.692) and *t* = −0.30 (*p* = 0.776); the corresponding funnel plots are shown in Figure 5. In female CS, *Z* = 0.37 (*p* = 0.707) and *t* = −1.39 (*p* = 0.235) and in male CS, *Z* = 1.56 (*p* = 0.117) and *t* = 1.26 (*p* = 0.247); the funnel plots for these are shown in Figure 5. Regarding suicidal ideation, in FS, *Z* = 1.04 (*p* = 0.296) and *t* = 1.22 (*p* = 0.436) and in CS, *Z* = 0 (*p* = 1) and *t* = −0.69 (*p* = 0.611); the funnel plots for these are shown in Appendix A. Regarding suicide attempts, in FS, *Z* = 0 (*p* = 1) and *t* = −0.02 (*p* = 0.981) and in CS, *Z* = 0 (*p* = 1) and *t* = −0.18 (*p* = 0.886); the funnel plots for these are shown in Appendix A. Regarding suicidal behaviours and exposure to tobacco, *Z* = 0.71 (*p* = 0.472) and *t* = 0.02 (*p* = 0.983); the corresponding funnel plots are shown in Figure 5.

These results indicate that there was no publication bias regarding the suicide risk of former, current, male, and female smokers because a conical distribution was maintained despite the dispersion, which was more likely attributable to heterogeneity. The results regarding ideation and suicide attempts were not reliable given the small number of articles included in the MA.

### 3.7. Sensitivity Analysis

Regarding deaths by suicide, the direction, significance, and precision did not significantly change in FS or CS with the elimination of each of the articles. In female CS, eliminating Lucas (2013) slightly reduced the risk (RR = 2.32) and precision (95% CI [1.75, 3.07]). In male CS, there were no major changes in the direction, significance, or precision, although eliminating Iwasaki (2005) did slightly increase the risk; Appendix A shows the corresponding influence graphs.

Regarding suicidal ideation, in FS, when Kang (2014) was eliminated, the risk was remarkably similar to that obtained globally (RR = 1.35) but the precision increased (95% CI [1.31, 1.39]), while eliminating any of the two other items increased the risk and decreased the accuracy. In CS, eliminating Kessler (2009) increased the risk (RR = 2.15) and precision (95% CI [2.09, 2.21]).

In terms of suicide attempts, in FS, eliminating Kessler increased the risk (RR = 1.84), eliminating Riala (2009) decreased the risk (RR = 0.89), and eliminating Berlin (2015) decreased the precision (95% CI [0.19, 8.34]). In CS, eliminating Kessler increased the risk (RR = 2.40), eliminating Riala (2009) decreased the risk (RR = 1.14), and eliminating Berlin (2015) decreased the precision (95% CI [0.31, 9.47]); the influence graphs are shown in Appendix A.

For suicidal behaviours and exposure to tobacco, there were no significant changes in direction, significance, or precision with the elimination of any of the articles; the influence graph is shown in Appendix A.

These results indicate that removing any of the included studies does not significantly change the results.

### 3.8. Subgroup Analysis

For deaths by suicide, both the *Q* index and the graph indicated heterogeneity in CS. Galbraith graphs (Appendix A) showed that the heterogeneity in both FS and CS was caused by Iwasaki (2005). By eliminating this article from the MA, we obtained an RR = 1.35 (95% CI [1.18, 1.54]) for FS and RR = 2.55 (95% CI [2.27, 2.86]) for CS. Given these results, no further subgroup analysis was necessary. In male CS, both the *Q* index and the graph indicated heterogeneity, which was caused by Iwasaki (2005) and Miller (2000b). By eliminating these two articles from the MA, we calculated a risk of suicide in male CS of RR = 2.05 (95% CI [1.65, 2.54]). Given the small number of articles on suicidal ideation and attempts, we were unable to perform subgroup analyses.

Regarding the risk of suicidal behaviours in participants exposed to tobacco, both the *Q* index and the Galbraith graph (Appendix A) indicated heterogeneity, with Clarke (2010) being the main article responsible for this variability. Eliminating the results from this article from the MA, we obtained a risk of suicidal behaviour of RR = 1.73 (95% CI [1.51, 1.99]), which is notably similar to that obtained previously. However, there was still heterogeneity (*Q* = 156.01; Galbraith graph in Appendix A), which mainly seemed to be caused by the differences between former and current smokers, and so these groups were best analysed separately, as carried out in the previous sections.

## 4. Discussion

In this present work we collected the evidence available to date from prospective studies that explored the relationship between smoking and suicidal behaviour, considering four dimensions: ideation, plans, attempts, and deaths by suicide. The results we obtained revealed that smokers presented almost double the risk for all these behaviours in every population type (including those with mental illnesses) and country considered. These data reinforce the hypothesis that smoking is an environmental risk factor for suicide, regardless of prior psychiatric or medical diagnoses or treatments [19]. This relationship decreased in FS, who had a lower RR than CS [19]; this fact was also reflected in our results, where FS had 1.3 times the risk and CS had a 2.4-fold increased risk compared to NS.

Similarly, other studies have also shown a dose–response association between CS and death by suicide, where an increase of 10 cigarettes/day was significantly associated with a 24% increase in the risk of suicide [11]. In addition, this seems to be a general relationship throughout the suicidal spectrum, given that previous studies indicated that the risk of developing suicidal ideation was 3.5-times higher in individuals who smoked 3.5 cigarettes per week [35], while in this current work the risk was 1.84 times higher in CS and only 1.35 times higher in FS.

The ‘gender paradox’ is traditionally used to explain the fact that there are more deaths by suicide among men compared to women, while the latter presented 3–4 times higher rates of ideation and suicide attempts [4]. According to our results, female smokers had a 2.5-fold higher risk of death by suicide than non-smokers and a higher RR than male smokers. In turn, male smokers had double the risk compared to NS. In contrast, male smokers were most at risk of suicide attempts with no result of death [36]. Thus, according to our results, the relationship between tobacco use and suicidal behaviours did not align with the gender paradox.

Multiple theories have been put forward to explain the biological basis of the relationship between tobacco use and suicide. On the one hand, researchers have considered the potential of nicotine to modify the central nervous system [37], associating smoking with a decrease in the enzymatic activity of MAO-A and B, thereby increasing the availability of the neurotransmitter MAO [38]. Genetic alterations in these enzymes have been related to greater impulsivity and loss of self-control, which are more frequent in smokers, and are known risk factors for suicide [16].

On the other hand, notable similarities become apparent if we extrapolate the monoaminergic hypothesis from depression to smoking. In the same way that the availability of serotonin is decreased in depressive conditions involving self-injurious behaviours [39], several post-mortem studies have shown that smoking could increase the risk of suicide by depleting serotonin levels in the hippocampus [40]. Smoking causes a decrease in oxygen levels (hypoxia) in most tissues, which several studies have concluded is related to an increased risk of suicide [41,42]. Likewise, there is another hypothesis that nicotine acts as a neurotoxin that can increase the risk of both depression and anxiety disorders [43]. It has also been hypothesized that tobacco use increases the risk of physical illness and that these organic disorders may increase the risk of suicide [44]. Furthermore, nicotine is a potent activator of the hypothalamic–pituitary–adrenal axis, and hyperactivity of this axis is a risk factor for suicidal behaviour [45,46].

Additionally, of note, addiction tends to affect people with a lower perception of danger, greater impulsivity, greater predisposition to take actions that endanger their life, and a tendency to act hastily without deliberation, planning, or supervision when faced with minimal stressors [47]. Constant sensation seeking is developed and cemented in these individuals through immediate positive and negative reinforcements, for example, immediate pleasure (positive reinforcer) or temporary regulation of negative emotions (negative reinforcer), despite potential long-term negative consequences [48]. This profile also has a greater tendency towards risk taking, which could favour less reflection when deciding to implement suicidal acts or gestures. In addition, smokers are exercising conscious harm against themselves with each cigarette consumed, thereby seeming to undervalue their health and life, perhaps also making them more likely to exercise self-injurious gestures. Finally, smoking as an addiction is associated with psychiatric comorbidities, which in turn increases the risk of suicide [49,50,51].

One limitation of this MA was the small number of studies we found on suicidal ideation, planning and attempts, which meant that we were unable to draw conclusions about the relationship between these factors and tobacco use. Likewise, only six studies presented data on women, which highlights the scarcity of gender perspectives in the work done to date. Another limitation was the mixing of heterogeneous results for the risk of suicidal behaviour in this MA. However, we considered that it would be useful to present a global measure of suicide risk related to tobacco exposure, even though the results referring to FS and CS separately were more reliable. This global measure reflects the importance of the relationship between smoking and suicidal behaviours. Moreover, it is important to note that, although all studies we included were prospective, the direction of causality should not be taken for granted because, as previously mentioned, multiple factors could mediate or cause the tobacco-suicide relationship. Thus, smoking can be considered as a contributing factor for suicide, although this association does not necessarily imply causation. It is also important to consider that the risk of suicide is not due to a single factor, but to the interaction between all risk factors.

Despite these limitations, our analyses of heterogeneity, sensitivity, and publication bias indicate that the relationship between tobacco use and the risk of death by suicide was robust. Thus, this MA suggested that the relationship between tobacco use and suicidal behaviour should be considered in clinical practice. Compared with other addictions, smoking is much more normalized, accepted, and accessible [52], and is less influenced by social desirability and concealment. In other words, a patient will be more likely to acknowledge that they use tobacco than disclose other risk factors related to suicide.

Prevention should begin by accurately evaluating suicide risk, which requires identification and evaluation of the main patient risk factors [4]. To this end, a wide variety of psychometric instruments have been proposed that complement and support clinical interviews. Hetero-administered scales are used in clinical practice, both in emergency departments and in primary and specialized care [4]. Among the most relevant to the context of this current work are the Beck Scale for Suicide Ideation (BSS), Beck Hopelessness Scale (BHS), Columbia Suicide Severity Rating Scale (C-SSRS), and SAD PERSONS scale [53]. The latter has been recommended because it is not difficult to remember and easy to apply [54], given that its name is a mnemonic device formed by the initials of the 10 items (suicide risk factors) that it evaluates: sex [male], age [<19 or >45 years], depression, previous attempts, excess alcohol abuse, rational thinking loss, social support [lacking], organized plan [for suicide], no partner or spouse, and sickness) [4,55].

However, none of the aforementioned scales include tobacco use as a risk factor. After analysing our results and considering the high prevalence of smoking and its impact on suicidal behaviours, we believe it would be appropriate to include tobacco use as a risk factor in suicide risk-assessment scales, especially given the simplicity of implementing this measure and its desirable cost–benefit ratio, both in terms of cost and time. For example, adding the ‘T’ (Tobacco) to the latter-mentioned scale would result in ‘T-SAD PERSONS’, which would also be easy for clinicians to remember. Given that the risk of suicide in smokers is similar to that found, for example, in single men (RR: 2.3) [4], which gives an idea of the importance of the relationship between smoking and suicide, it is pertinent to study whether the inclusion of tobacco in the scales improves the detection of individuals at risk. Although suicide risk scales have a high rate of false positives, it is worth it if we reduce the number of false negatives.

Furthermore, the results of this current work indicate that more studies have been carried out on death by suicide than on the other three dependent variables on the suicide spectrum (suicide ideation, planning, and attempts). This fact may be due to the existence of official national registries that record suicide deaths, while it is more difficult to objectify and numerically record data about the other behaviours. Therefore, it should be emphasized that the variables of self-injurious ideation, plans for suicide, and previous suicide attempts must be ruled out in clinical healthcare settings, regardless of the reason for the patient consultation. To achieve this, an ideal therapeutic environment must be created in which patients have a good relationship with their healthcare provider (in order to promote good communication and a more individualized approach) and where healthcare providers would receive specialized training by mental health specialists [56]. It would also improve our knowledge on this topic if the studies would further detail the ages of the participants to check if the relationship between smoking and suicide varies with age and always include data on men and women.

Finally, considering that suicide is the leading global cause of violent deaths, it is vitally important to assess the risk of suicidal spectrum behaviours in CS patients (especially women), as well as in FS, both through clinical interviews and by using clinical scales. Thus, suicide evaluation protocols must be standardised, and their use emphasised alongside a National Suicide Prevention Plan. Indeed, many nations, including Spain, are yet to establish and implement such plans, despite the recommendations of the WHO and EU [57]. In that sense, our study shows how the community would benefit from these prevention guidelines, taking in account suicide risk factors, such as tobacco. Therefore, we would have an argument in favour of the usefulness of generating new public health measures for the reduction/elimination of smoking, which could be a promising way to reduce the risk of suicidal spectrum behaviours promoting health.

## 5. Conclusions

The risk of death by suicide is 2.5-times higher among smokers. Men who smoke have twice the risk of suicide, while female smokers have an even higher risk (a 2.5-fold increase). Exposure to tobacco, whether present or past, almost doubles the risk of presenting some behaviour on the suicidal spectrum (ideation, plans, attempts, or death by suicide). Thus, the systematic evaluation of tobacco use should be included in every patient assessment (particularly in individuals at an apparent risk of suicide) and the treatment of smoking ought to be part of suicide prevention strategies.

## Figures and Tables

**Figure 1 ijerph-18-06103-f001:**
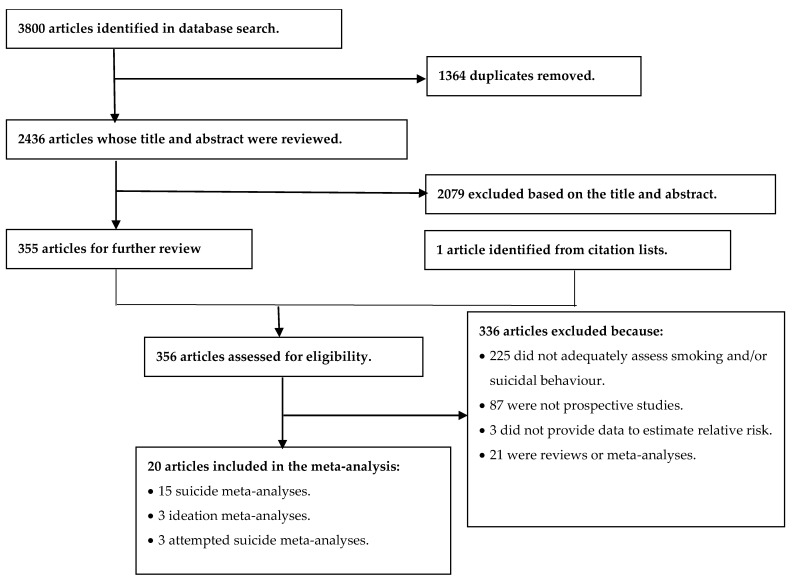
Flowchart of the systematic review process used to select articles for inclusion in this meta-analysis.

**Figure 2 ijerph-18-06103-f002:**
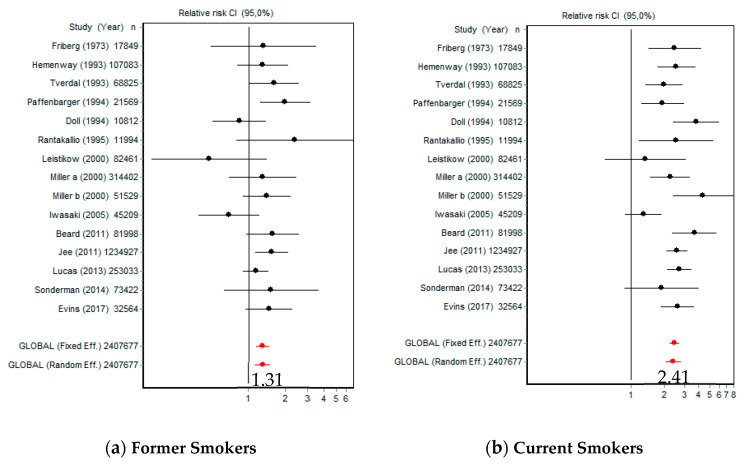
Forest plots of reported suicide rates in former and current smokers.

**Figure 3 ijerph-18-06103-f003:**
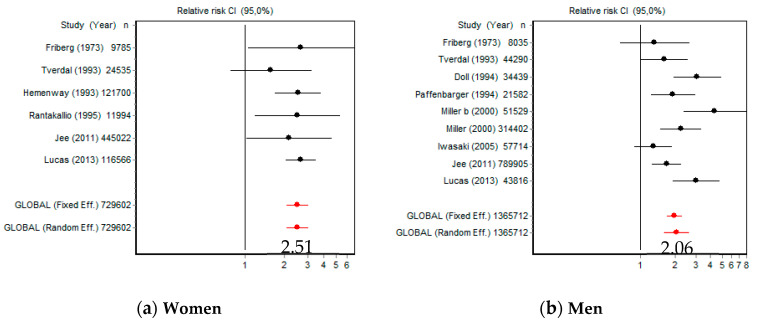
Forest plots of reported suicide rates in female and male smokers.

**Figure 4 ijerph-18-06103-f004:**
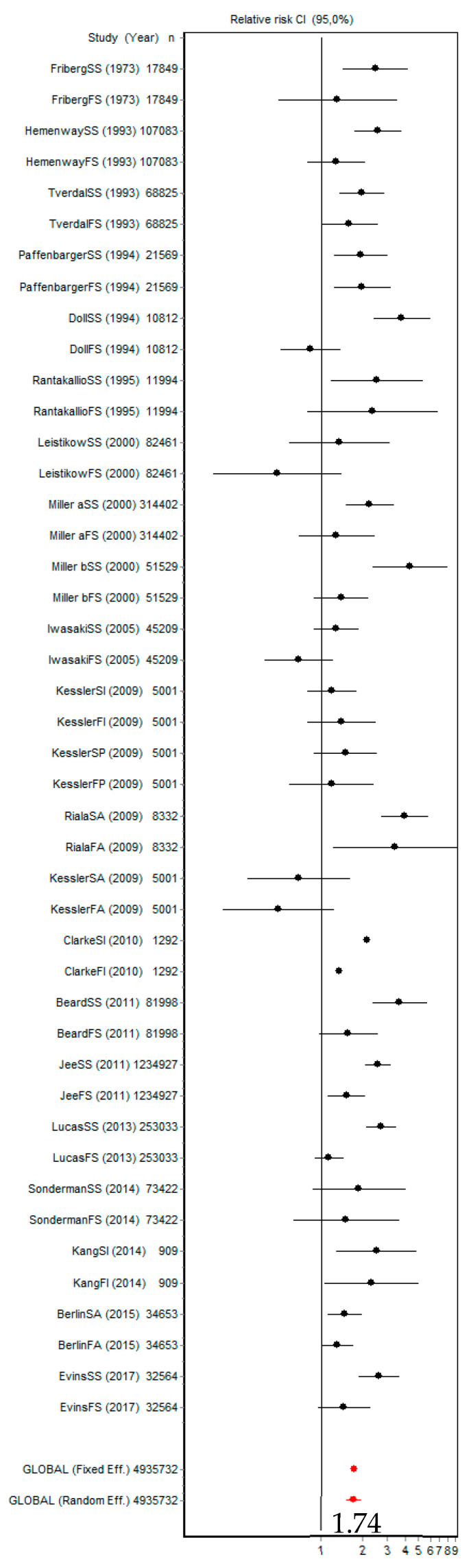
Forest plot for all the reported suicidal behaviours (ideation, plans, attempts, and death by suicide) in former and current smokers.

**Figure 5 ijerph-18-06103-f005:**
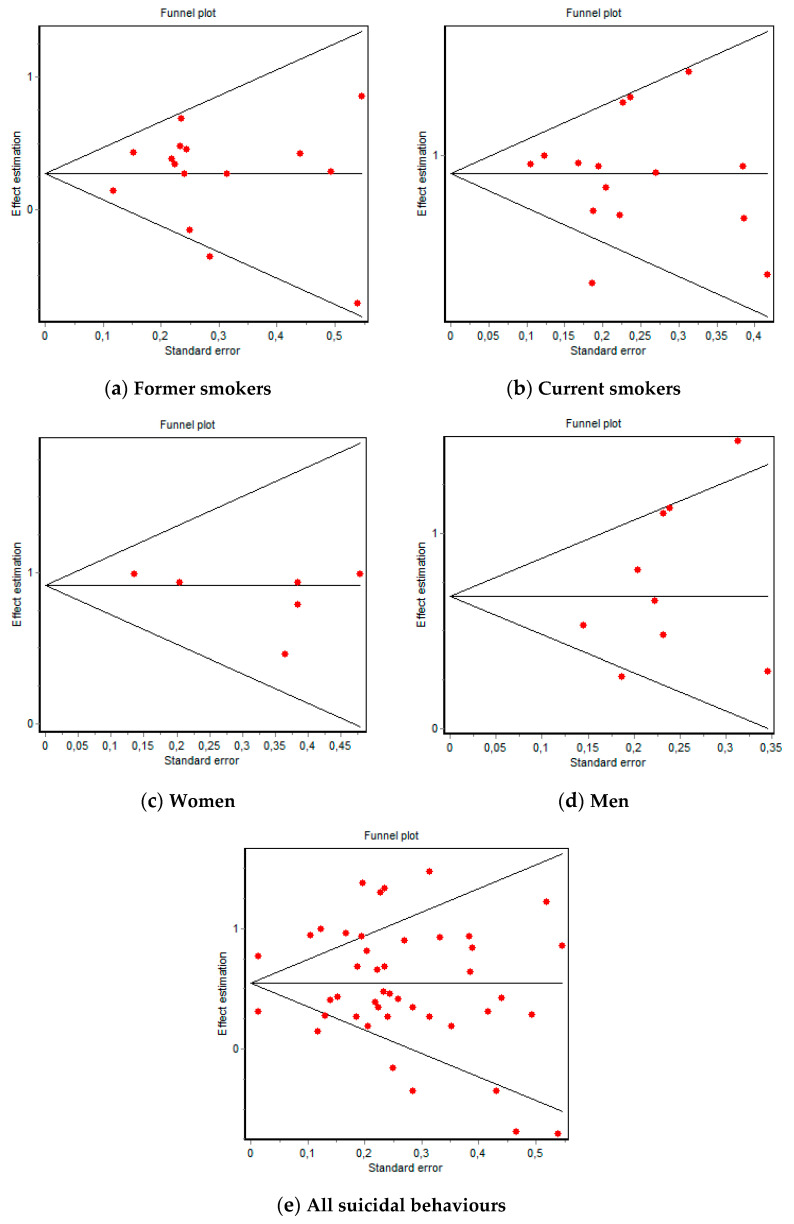
Funnel plots of suicide and all suicidal behaviours (ideation, plans, attempts, and death by suicide).

**Table 1 ijerph-18-06103-t001:** Studies included [15,16,17,18,19,20,21,22,23,24,25,26,27,28,29,30,31,32,33,34].

1st Author Year	Country	Population	Sex	Age	Follow-Up (Years)	Sample	Suicidal Behaviours	NOS Quality
Beard 2011	USA	Pesticide applicators	Both	>18	13.3	81,998	Suicide	8
Berlin 2015	USA	General	Both	>18	3.5	34,653	Attempt	9
Clarke 2010	USA	General	Both	18–54	12	1292	Ideation	9
Doll 1994	Britain	Doctors	Male	>35	40	10,812	Suicide	5
Evins 2017	Finland	Twins	Both	>18	6	32,564	Suicide	8
Friberg 1973	Switzerland	Twins	Both	>36	11	17,849	Suicide	7
Hemenway 1993	USA	Nurses	Female	30–55	12	107,083	Suicide	6
Iwasaki 2005	Japan	General	Male	>40	10	45,209	Suicide	8
Jee 2011	Korea	General	Both	30–95	14	1,234,927	Suicide	8
Kang 2014	Korea	General	Both	>65	2.4	909	Ideation	9
Kessler 2009	USA	General	Both	15–54	13	5001	I/P/A	9
Leistikow 2000	USA	General	Both	>18	5	82,461	Suicide	9
Lucas 2013	USA	Health professionals	Both	>25	32	253,033	Suicide	7
Miller 2000a	USA	Army soldiers	Male	>17	10	314,402	Suicide	7
Miller 2000b	USA	Health professionals	Male	40–75	8	51,529	Suicide	7
Paffenbarger 1994	USA	Harvard alumni	Male	35–74	27	21,569	Suicide	6
Rantakallio 1995	Finland	Pregnant women	Female	14–49	28	11,994	Suicide	7
Riala 2009	Finland	General	Both	14–31	17	8332	Attempt	8
Sonderman 2014	USA	General	Both	40–79	9	73,422	Suicide	8
Tverdal 1993	Norway	General	Both	35–49	13.3	68,825	Suicide	9

Note: I: Ideation; P: Plans; A: Attempts.

## Data Availability

Data from this study are available from the corresponding author on request.

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
