# Peer review of "Proposal for the Inclusion of Tobacco Use in Suicide Risk Scales: Results of a Meta-Analysis"

_ijerph, 2021, doi:10.3390/ijerph18116103_

Round 1

Reviewer 1 Report

This serves as a review for "Proposal for the inclusion of tobacco use in suicide risk scales: Results of a meta-analysis." Overall - I think the article is well written and I commend the authors on their good work. Below I've outlined a few minor points to be considered.

(1) The authors mention the statistical results - but do not provide any context on what these statistical results mean. They do a good job of this later in the results - in the "Publication Bias" section they have a paragraph starting with "These results indicate..." This is a good example of how I think they should handle all the results subsections. 

(2) In the discussion the authors state, "were more at risk of unsuccessful suicide attempts" - I would avoid use of successful or unsuccessful in describing suicidal behavior. 

(3) One thing that might be useful to address in the discussion is the fact that suicide risk assessment is plagued by issues of false positives. This seems particularly important given the fact that the authors note that there are 1,337 million people worldwide who smoke compared to 800,000 suicide deaths annually across the globe. Only a small number of those who smoke will go on to die by suicide - what is the likelihood that weighing smoking behavior into risk assessment is actually going to improve our ability to identify those at risk of suicide? 

Author Response

1-We have included sentences explaining the meaning of the results in the rest of the subsections:

'In FS, both the Q index and the Galbraith graph indicate the absence of heterogeneity. In CS, the Q index indicates the presence of heterogeneity, but in the graph it can be seen that only Iwasaki (2005) contributes to it. In women, both the Q index and the Galbraith graph indicate absence of heterogeneity. In men, the Q index indicates heterogeneity, but in the graph it can be seen that Miller (2000b) is the one that contributes the most to this heterogeneity'.

'In FS, both the Q index and the Galbraith graph indicate the absence of heterogeneity. In CS, the Q index indicates the presence of heterogeneity, but in the graph it can be seen that such heterogeneity is minimal'.

'In FS, Q index indicates heterogeneity, but the Galbraith graph indicates the absence of heterogeneity. In CS, both the Q index and the graph indicate the presence of heterogeneity'.

'Both the Q index and the graph indicate the presence of heterogeneity'.

'These results indicate that removing any of the included studies does not significantly change the results'.

2-We agree that is an unfortunate expression. We have replaced it with suicide attempts with no result of death.

3-We have added in the discussion: Given that the risk of suicide in smokers is similar to that found, for example, in single men (RR: 2.3) [4], it is pertinent to study whether the inclusion of tobacco in the scales improves the detection of individuals at risk. Although suicide risk scales have a high rate of false positives, it is worth it if we reduce the number of false negatives.

We greatly appreciate your suggestions, they have helped us to improve the article. Thank you very much. 

Reviewer 2 Report

In my opinion, the study is very interesting and worth being published. In fact, the article is well-structured, the method used is congruent with the purpose of the study and the results are presented clearly.  

Nevertheless, the Background and Findings sections are not sufficiently developed to exhibit the value of the research undertaken by the authors. For this reason, I feel I can give the following suggestions:

  1. Background – Expand a little more to highlight the research problem to highlight the study's need.
  2. Findings: It would be a good paper if it did look at the research impact on the community. Expand a little more.
  3. In the list of references, the name of the journal is not abbreviated, check them all.

I think the authors can easily follow the suggestions I have given in this review and make a new version of their interesting paper.

All best wishes.

Author Response

1-We have added in the introduction: Previous meta-analyses conclude that smoking is associated with a higher risk of completed suicide [14]. However, we do not know how smoking could influence on the rest of the suicidal spectrum behaviours, since we have not found any prospective meta-analyses that analyses each of these behaviors separately. In case of concluding that such association exists, we would be closer to considering smoking as a modifiable risk factor for suicide.

2-We have added in the discusión: In that sense, our study shows how community would benefit from these prevention guidelines, taking in account suicide risk factors, such as tobacco. Therefore, we would have an argument in favor of the usefulness of generating new public health measures, such as legislation for the reduction / elimination of smoking, which could be a promising way to reduce the risk of suicidal spectrum behaviours and promote health.

3-We have checked and modified the following:

37.Benowitz, N.L. Nicotine addiction. N Engl J Med 2010, 362, 2295–303. Doi: 10.1056/NEJMra0809890.

55.Rangel-Garzón, C.X.; Suárez-Beltrán, M.F.; Escobar-Córdoba, F. Escalas de evaluación de riesgo suicida en atención primaria. Rev Fac Med 2015, 63(4), 707-716. Doi: 10.15446/revfacmed.v63.n4.50849.

We greatly appreciate your suggestions, they have helped us to improve the article. Thank you very much.

Reviewer 3 Report

The article “Proposal for the Inclusion of Tobacco Use in Suicide Risk Scales: Results of A Meta-analysis” proposes the inclusion of a little analyzed risk factor such as tobacco use and its relationship with suicidal behavior. It is important to include various risk factors in the existing scales, especially in primary care, to allow a broader exploration as well as more targeted and effective interventions for the entire population.

Below, there are some suggestions for improvement of the present study:

  • It would be interesting to emphasize throughout the study the importance that smoking is associated with an increased risk of suicidal behaviors. Smoking can be considered as a contributing factor for suicide, although this association does not necessarily imply causation.
  • It is difficult to focus the subject of the study on a single factor such as tobacco use. Throughout the text, it should be emphasized that the risk increases with the sum of the risk factors and not just the occurrence of one of them.
  • Has age heterogeneity been taken into account in the analysis of results? Is there a greater risk when consumption occurs at a younger age?
  • Add numerical data to Figures 2, 3 and 4 to facilitate the reading and understanding of the data represented.
  • I propose to add in the introduction data on the different risk factors analyzed in suicidal behavior to emphasize the importance of adding "tobacco" as one more. For example:

Ribeiro, J. D., Franklin, J. C., Fox, K. R., Bentley, K. H., Kleiman, E. M., Chang, B. P., & Nock, M. K. (2016). Self-injurious thoughts and behaviors as risk factors for future suicide ideation, attempts, and death: a meta-analysis of longitudinal studies. Psychological medicine46(2), 225-236.

  • I suggest adding in the introductory part information regarding the different expressions of suicidal behavior to discern if there is relevance as to when some risk factors are more important than others.
  • It is surprising that no information regarding current and relevant studies, such as, for example, the following, has been included in this study:

Tanskanen, A., Tuomilehto, J., Viinamäki, H., Vartiainen, E., Lehtonen, J., & Puska, P. (2000). Smoking and the risk of suicide. Acta Psychiatrica Scandinavica101(3), 243-245.

Hughes, J. R. (2008). Smoking and suicide: a brief overview. Drug and alcohol dependence98(3), 169-178.

  • “Considering that numerous studies suggest that tobacco use increases suicide risk [6]” Add more studies if they are claimed to be numerous (as the previous ones proposed) or modify the sentence.

Author Response

1-We have modified the limitations paragraph leaving it like this: ‘This global measure reflects the importance of the relationship between smoking and suicidal behaviours. Moreover, it is important to note that, although all the studies we included were prospective, the direction of causality should not be taken for granted because, as previously mentioned, multiple factors could mediate or cause the tobacco-suicide relationship. Thus, smoking can be considered as a contributing factor for suicide, although this association does not necessarily imply causation’.

We have added in the discusión ‘Given that the risk of suicide in smokers is similar to that found, for example, in single men (RR: 2.3) [4], which gives an idea of the importance of the relationship between smoking and suicide, it is pertinent to study whether the inclusion of tobacco in the scales improves the detection of individuals at risk.

2-We have added to the limitations: ‘It is also important to consider that the risk of suicide is not due to a single factor, but to the interaction between all risk factors’.

3-This is a very interesting question. Most of the articles did not include the mean age, while, as can be seen in Table 1, many did not give an age range, but rather from what age participants were recruited. This makes very difficult to group them with guarantees by age to perform the heterogeneity analysis. However, if we take as a reference the ages that increase the risk of suicide (<19 and> 45), we see in the Galbraith graphs that neither the study with the oldest sample (Kang) nor those that include the youngest subjects (Kessler, Rantakallio, and Riala) contribute to heterogeneity in suicide risk. In fact, eliminating these studies from the meta-analysis does not modify the results obtained regarding completed suicide.

However, it is a pity not to be able to carry out this analysis adequately due to the lack of good data in this regard. For this reason, we have added in the discussion: it would also improve our knowledge on this topic if the studies would further detail the ages of the participants to check if the relationship between smoking and suicide varies with age and always include data on men and women.

4-We have added the numerical value of the pooled risk ratio.

5-We have added in the introduction: Many of the risk factors for suicide are also risk factors for being a smoker: being young, non-white, lower income, less education, unmarried, unemployed, less religious, anxiety, depression, psychoses, substance use problems, low self-esteem, risk taking, having a serious physical illness, impulsivity, aggression, antisocial personality [7].

6-We have added in the introduction: Previous suicide ideation was the most robust predictor of later ideation. Prior non-suicidal self-injuries or suicide attempt conferred the most risk for later suicide attempts and previous suicide attempt and suicide ideation were among the strongest predictors of suicide death [7].

7-We have included these studies in our paper in the references 7-9. 

8-We have added several articles that find this relationship in the references 6-9.

We greatly appreciate your suggestions, they have helped us to improve the article. Thank you very much.